# An Exploratory Study of Influenza Vaccination Coverage in Healthcare Workers in a Western Chinese City, 2018–2019: Improving Target Population Coverage Based on Policy Interventions

**DOI:** 10.3390/vaccines8010092

**Published:** 2020-02-19

**Authors:** Lili Xu, Jinhua Zhao, Zhibin Peng, Xiaojin Ding, Yonghong Li, Huayi Zhang, Huaxiang Feng, Jiandong Zheng, Hailan Cao, Binzhong Ma, Yan Shi, Yongcheng Ma, Luzhao Feng

**Affiliations:** 1Institute for Infectious Disease Control and Prevention, Qinghai Provincial Center for Disease Control and Prevention, Qinghai 810007, China; qhcdcxull@126.com (L.X.); qhcdczjh@126.com (J.Z.); qhcdcdxj@126.com (X.D.); qhcdclyh@126.com (Y.L.); qhcdczhy@126.com (H.Z.); qhcdcfhx@126.com (H.F.); qhcdcchl@126.com (H.C.); qhcdcmbz@126.com (B.M.); qhcdcsy@126.com (Y.S.); 2Key Laboratory of Surveillance and Early-warning on Infectious Disease, Division of Infectious Disease, Chinese Center for Disease Control and Prevention, Beijing 102206, China; pengzb@chinacdc.cn (Z.P.); zhengjd@chinacdc.cn (J.Z.)

**Keywords:** influenza, healthcare workers, vaccination coverage, policy intervention, Xining, China

## Abstract

**Objectives:** To evaluate a policy-based intervention to increase seasonal-influenza-vaccination coverage in healthcare workers in Xining, a city in Western China. **Methods:** From October 2018 to March 2019, we implemented a free vaccination policy in healthcare workers in Xining. A face-to-face interview with the head of the infection control department and an online survey for medical staff in four tertiary medical facilities was conducted to understand both the implementation of the free policy and influenza vaccination coverage. Possible factors for influenza vaccination among healthcare workers (physician, nurses working on the front-line, HCWs) were investigated by multivariate-logistic regression. **Results:** Coverage in two hospitals that implemented the free vaccination policy was 30.5% and 25.9%, respectively, which was statistically different to hospitals that did not implement the free policy (7.2% and 8.7%, respectively) (χ2 = 332.56, *p* < 0.0001). Among vaccinated healthcare workers, 65.5% and 48.6% reported their main reasons for vaccination were a convenient vaccination service and awareness of the free vaccination policy. The reasons for not being vaccinated among the 3389 unvaccinated healthcare workers included: the inconvenient vaccination service (33.8%), believing vaccination was unnecessary (29.7%), concerns about adverse reactions to the vaccine (28.8%), and having to pay for the vaccine (25.6%). **Conclusions:** Implementing the free vaccination policy, combined with improving the accessibility of the vaccination service, increased seasonal-influenza vaccination-coverage in healthcare workers in Xining.

## 1. Introduction

Influenza is an acute respiratory infectious disease caused by the influenza virus, and leads to between 290,000 and 650,000 deaths globally every year [1]. Healthcare workers (HCWs) are at high risk of influenza infection because of their occupational exposure to infected patients. A systematic review has shown that healthcare workers who are not vaccinated have a 3.4 times higher risk of developing influenza than healthy adults [2]. In addition to morbidity among healthcare workers, influenza infection may also lead to increased absenteeism, presenteeism and disruption of medical services [3,4]. Moreover, although hard to quantify the risk, healthcare workers infected with influenza, regardless of whether they have influenza-like symptoms, may contribute to the nosocomial transmission of infection to their patients. It can lead to the occurrence of severe influenza and complications, especially in high-risk groups. Previous studies have also shown that a high percentage of healthcare workers infected with influenza continue to work after developing influenza-like symptoms [5].

When the influenza vaccine strains were well-matched with the epidemic strains, vaccine efficacy for healthcare workers reached 90% [6,7], suggesting that receiving the vaccine would significantly reduce morbidity and absenteeism among healthcare workers. A systematic review found that vaccination in healthcare workers was effective against laboratory-confirmed influenza, with a corresponding decrease in absenteeism due to influenza-like illness and a decrease in the length of absence [8]. Further, influenza vaccination of healthcare workers contributes to influenza pandemic preparedness [9]. Moreover, studies found that vaccinated healthcare workers are more likely to recommend vaccination to their patients and could greatly increase public uptake [10,11].

More than 90 countries around the world recommend the influenza vaccination for healthcare workers. The technical guidelines for seasonal influenza vaccination in China (2018–2019)also considers healthcare workers to be a priority target group. Information on influenza vaccination coverage in healthcare workers was not available for most countries, though it varied widely and was reported to be low [9]. An Internet survey of 2000 healthcare workers in the United States showed that the influenza vaccination coverage was 78% in the 2017–2018 season and up to 95% among those working in settings where vaccination was required by employers [12]. In China, according to the results of a 2018 systematic review, the highest vaccination coverage rate among healthcare workers in five epidemic seasons since 2010 was no more than 15% [13]. A few small-scale surveys in some cities, such as Ningbo [14], Qingdao [15], and Xining [16], have shown that the vaccination coverage ranged from 5.4% to 12.4% during the recent flu season.

## 2. Materials and Methods

### 2.1. Study Area

The city of Xining, the capital of the Qinghai province located in the northwest of China, has a population of 2.37 million residents and 31,500 healthcare workers registered in 2018, accounting for about 61% of the total number of healthcare workers in the whole province. More than two-thirds of the region’s tertiary hospitals are located in Xining, and residents are more willing to go to these medical institutions to tackle health problems. Typically, there is only one winter–spring seasonal influenza peak, during November to March, in Xining each year, which is in line with most northern Chinese cities. As the influenza vaccination is a category 2 vaccine, it was voluntary and an out-of-pocket expense, with no local free vaccination policy until 2017.

Before the 2017–2018 influenza season, Qinghai CDC conducted a knowledge, attitude and practice interview regarding seasonal influenza and influenza vaccination among healthcare workers from four randomly selected tertiary hospitals in Xining City. The result showed that the coverage rate of influenza vaccines among the clinical staff was 5.14% (95% *CI*, 4.80–5.49%) during the 2016–2017 influenza season. The main reasons for not being vaccinated included not having time to be vaccinated (52.34%) and thinking it was unnecessary to be vaccinated (36.42%). Furthermore, the proportion of those who recommended the influenza vaccination to others was 88% in the vaccinated group, which was statistically higher than the 39% in the unvaccinated group (χ^2^ = 99.57, *p* < 0.0001) [16].

### 2.2. Methods and Content

In 2017, the government of Qinghai introduced a free influenza vaccinations policy for government cadres and staff, including the medical staff. However, it had not been implemented widely because of a shortage of vaccines during the 2017–2018 season. From October 2018 to March 2019, the hospitals in Xining City were required to perform centralized or temporary vaccinations for free among healthcare workers before the epidemic season. The four hospitals mentioned above were selected to implement and evaluate the free policy interventions to increase seasonal-influenza-vaccination coverage among healthcare workers. A face-to-face interview with the head of the infection control department of the hospitals and online surveys for medical staff were conducted to understand both the implementation of the free policy and influenza vaccination coverage in May 2019.

It was discovered from the interview whether the hospital implemented the free policy and provided temporary or centralized vaccination services on the hospital grounds. Details were obtained as to whether the vaccination site was located in the hospital’s clinic and when was vaccination available for HCWs. Demographic information, and knowledge about influenza, as well as the vaccination and the vaccination status during the 2018–2019 season were investigated using a self-administered questionnaire. Data about healthcare workers’ willingness to be vaccinated during the 2019–2020 season, and factors associated with vaccination and not being vaccinated, were also collected.

### 2.3. Statistical Analysis

Descriptive statistical methods were used to calculate vaccination coverage and the number of respondents based on reasons for and against vaccination. Categorical variables were compared using the Pearson chi-squared test. All statistical tests were two-sided and considered significant at a level of 0.05 using SPSS 22.0 software.

Possible factors for obtaining the influenza vaccination among HCWs were investigated by multivariate-logistic regression, with HCWs vaccination status during the 2018–2019 season as a dependent variable. All variables with *p* values < 0.10 in the univariate analysis were selected for multiple-logistic-regression analysis (forward-stepwise-regression algorithms) as independent variables, including occupation, gender, age, years of medical service, working department, average number of patients served per day, whether the hospital provides the free vaccine and whether the hospital provides a vaccination service.

### 2.4. Ethics Approval

The study protocol and questionnaire were approved by the Ethical Review Committee of Chinese Center for Disease Control and Prevention (No. 201901, China CDC, Beijing, China). All participants provided verbal informed consent to be interviewed.

## 3. Results

In total, 4094 valid questionnaires were received from four tertiary hospitals with a response rate of 64.3%. Nearly 83% of the respondents were 20–39 years of age, with the rest over 40 years of age or under 20 years of age. There were 595 males and 3499 females, and the ratio of male to female was approximately 0.17:1. The number of years in medical service was 28.7% for 6–10 years, followed by 3–5 years (26.9%) and 11–20 years (19.9%). The number of people who worked in pediatrics, the infectious diseases department, emergency department and the respiratory department was 1200, accounting for 29.3%.

The coverage rates for the seasonal influenza vaccination in healthcare workers were 7.2% (95 CI: 5.9–8.4%) and 8.7% (95 CI: 6.6–10.8%) in the two hospitals that did not implement free vaccinations. At the same time, the coverage rates of the two hospitals that implemented the free vaccination policy and provided centralized vaccination or temporary services were 25.9% (95 CI: 22.3–29.4%) and 30.5% (95 CI: 28.0–33.1%), respectively. The coverage of HCWs showed significant differences in these four different hospitals (χ^2^ = 332.56, *p* < 0.0001). Compared with the previous season, coverage among HCWs in the two hospitals that implemented the free vaccination policy increased significantly (χ^2^ = 61.3 and 256.65, *p* < 0.0001) (Figure 1).

Out of 705 respondents who were vaccinated during the 2018–2019 season, the most common reason for being vaccinated was that the hospitals provided centralized or temporary vaccinations, accounting for 65.5% (462/705) of respondents, followed by the concern of developing influenza (51.3%, 362/705). The third reason was “the free vaccination policy”, which accounted for 48.6% (343/705). The reasons for not being vaccinated among the 3389 unvaccinated HCWs included: an inconvenient vaccination service (33.8%), believing vaccination was unnecessary (29.7%), concerns about adverse reactions from the vaccine (28.8%), and vaccines not being free (25.6%). Out of the unvaccinated HCWs who indicated an “inconvenient vaccination service” and “vaccination at their own expense”, willingness to be vaccinated was over 85% should the vaccination become free during the 2019–2020 season (Figure 2).

Table 1 shows the result of multiple-logistic-regression models on factors affecting influenza vaccination among healthcare workers in Xining City in the 2018 to 2019 season. Female medical staff; people whose medical service >20 years; people whose average number of patients served >200 per day; and those working in high-risk departments including respiration, infectious diseases, emergency, pediatrics, ICU/NICU, were more likely to accept the vaccines. The results show that HCWs who were provided with free vaccines and vaccination services in hospitals was 8.80 times (95% CI: 6.43–12.05) and 1.71 (95% CI: 1.20–2.43) times more likely to be vaccinated than those who were not provided free vaccination and services.

More than 90% of HCWs believed that influenza can cause further complications that can lead to severe illness and death, and believed that they themselves are at a high risk of infection and could transmit the virus to others. The percentages of HCWs who believed infants between 6 and 23 months of age, pregnant women and women trying to become pregnant during the influenza season were the priority for vaccination were 43.4% and 37.7%, respectively. There was no significant difference between the vaccinated and unvaccinated groups for these questions. There were differences between the vaccinated group and the unvaccinated group regarding the perception of “the influenza vaccination as the most effective measure to prevent influenza”, “the optimal time of vaccination”, and “HCWs as the priority for vaccination”, as shown in Table 2. Meanwhile, the percentage of vaccinated HCWs in the 2018 to 2019 season who recommended the influenza vaccination to patients was 68.7% (484/705), significantly higher than the 34.9% (1183/3389) in the unvaccinated group (χ^2^ = 257.32, *p* < 0.0001).

## 4. Discussion

Complete implementation of the free vaccination policy, combined with improving the convenience and accessibility of the vaccination service, significantly increased seasonal-influenza vaccination-coverage in healthcare workers in Xining. The reasons for vaccination, such as the “hospital provided centralized or temporary vaccinations” and a “free vaccination policy” were in first and third place, respectively. However, the free vaccinations for HCWs in Xining City did not cover the extra staff in hospitals currently in the program. There are still some obstacles to vaccination among HCWs including the implementation of special funds for this target group in the hospitals, inadequate knowledge of influenza vaccines and doubts about the vaccine among professionals.

Coverage rates of the seasonal influenza vaccination in HCWs vary widely in different countries and regions. European countries during the 2014–2015 influenza season reported that 26 (56%) countries provided influenza vaccination data for healthcare workers, with coverage rates ranging from 2.6% to 99.5% [17]. In the three seasons from 2015 to 2018, coverage rates among HCWs in the United States remained around 78% to 79% [12]. The coverage of the 2016–2017 and 2017–2018 seasons in Xining City were 5.1% and 6.8%, respectively [16,18], according to our surveys. A range of beliefs may prevent vaccination, including concerns about side effects, doubts about vaccine efficacy, a lack of time during their working hours and inconvenient services, leading to low vaccination coverage among healthcare workers in China.

Previous studies have also found that cost is a common obstacle to obtaining an influenza vaccination, especially in places where vaccination is not covered or subsidized by insurance [19,20]. However, the influenza vaccination was free for HCWs in some regions of China, but the number of participants was limited. In our study, 25.6% of HCWs did not know that they could be vaccinated for free and the vast majority would have been vaccinated had they known. This indicated that a free policy alone could not improve vaccination if accessibility services and policy are not widely known and implemented [18]. The findings of the survey showed that the most common reason for vaccination was that the hospitals provided centralized or temporary vaccinations, whilst, reasons for not being vaccinated mainly included an inconvenient vaccination service.

A large number of studies, consistent with our findings, have shown that measures such as providing free vaccinations, improving accessibility to vaccination services and actively promoting the vaccination can significantly improve the vaccination rate in medical staff [21,22]. However, in the absence of legislative or mandatory vaccination requirements, it is difficult to reach a high level of coverage in HCWs [23,24]. In “the notice on further strengthening the prevention and control of influenza (2018)”, the Chinese National Health Commission explicitly requires medical institutions at all levels to provide free influenza vaccination services to their medical staff. Therefore, local governments and medical facilities need to explore comprehensive and effective interventions that include policy and service to improve influenza vaccination coverage among healthcare workers [25].

Many previous studies have found that the combination of an educational and a promotional element appeared the most effective in improving influenza vaccination among healthcare workers [26]. More than 90% of HCWs, both vaccinated and unvaccinated, understood the dangers of influenza and their own occupational risks. However, the survey found people whose medical service was longer than 20 years and those working in high-risk departments were more likely to accept the vaccines. It indicated that the awareness of influenza and the influenza vaccine can improve vaccination in HCWs. Further, we found that there were differences in the perception of the influenza vaccination and in recommending it between the two groups. Healthcare workers in the vaccinated group were more likely to recommend the influenza vaccination to their patients, as indicated in previous studies [13,16]. Therefore, we recommend improving the awareness and coverage rate of influenza vaccinations in HCWs by developing clinical guidelines and expert consensus. The increased awareness and vaccination of healthcare workers could further encourage clinicians to recommend influenza vaccination, to achieve the target of improving vaccination rates in groups at high risk of influenza in China.

This study has at least three limitations. First, the results regarding the vaccinations were self-reported and vaccination records were not verified further. We think healthcare workers are mostly professional, and the possibility of the false reports was, therefore, relatively low. Second, with cluster sampling and a response rate of 64.3%, it may not be identical to all the basic components of the original group. Third, the two hospitals that implemented the free policy also provided centralized or temporary vaccination services, and the impact of the free policy and vaccination service could not be evaluated separately.

## 5. Conclusions

Vaccinating healthcare workers not only protects them, their patients and families, but also keeps healthcare facilities functioning during an influenza epidemic. Free vaccination policies combined with accessibility services lead to an evident improvement in influenza vaccination coverage in the healthcare workers surveyed in our study. Meanwhile, our study suggests that an increased vaccination rate in HCWs has been accompanied by an increase in vaccination recommendations to their patients. Therefore, we should continue to explore effective interventions to increase the coverage rate of medical workers in order to promote the vaccination of high-risk groups and even general populations.

## Figures and Tables

**Figure 1 vaccines-08-00092-f001:**
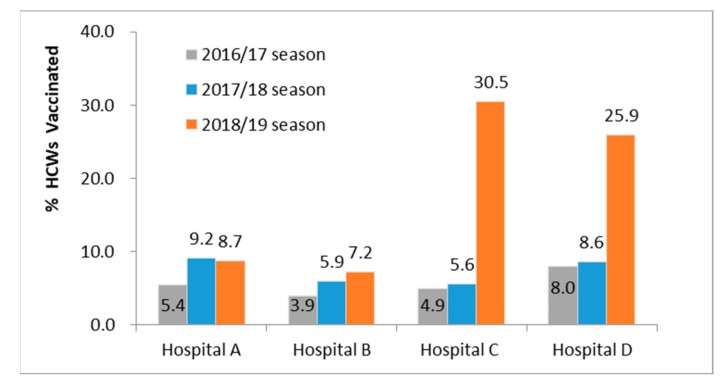
Seasonal-influenza-vaccination coverage in healthcare workers in two hospitals providing free vaccinations and two control hospitals, Xining, China, 2016–2019. Hospitals C and D implemented a free vaccination policy in the 2018–2019 season; hospitals A and B did not have the free policy.

**Figure 2 vaccines-08-00092-f002:**
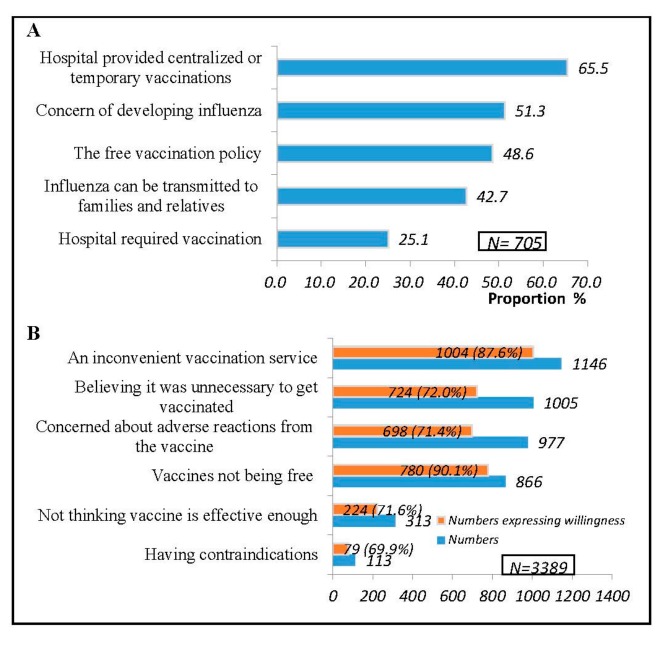
Reasons for receiving and not receiving the influenza vaccination and willingness to be vaccinated among healthcare workers during the 2019–2020 season. **Panel A**: Reasons for being vaccinated and response rate; **Panel B**: Reasons for not being vaccinated and willingness to be vaccinated if vaccination became free in the hospital during the 2019–2020 season.

**Table 1 vaccines-08-00092-t001:** The multiple-logistic regression results of factors associated with influenza vaccination of healthcare workers in Xining City, Qinghai, China.

Variables	*β*	S_X_(*β*)	*p* Value	OR	OR 95% CI
Gender (Male)	−0.36	0.14	0.01	0.70	0.54–0.91
Years of medical service ^a^					
>20 Years	0.88	0.14	< 0.001	2.42	1.83–3.19
Working department					
High-risk department ^b^	1.43	0.10	< 0.001	4.18	3.45–5.06
Average number of patients served per day ^c^					
>200 patients per day	0.85	0.27	0.002	2.33	1.38–3.95
Hospital provides free vaccine	2.18	0.16	<0.001	8.80	6.43–12.05
Hospital provides vaccination service	0.54	0.18	0.003	1.71	1.20–2.43

^a^ Setting dummy variable to years of medical service, <5 years as a reference group. ^b^ High-risk departments include respiration, infectious diseases, emergency, pediatrics, ICU/NICU. ^c^ Setting dummy variable to average number of patients served per day, <50 patients as a reference group.

**Table 2 vaccines-08-00092-t002:** Knowledge about the influenza vaccination among vaccinated and unvaccinated healthcare workers in Xining City, Qinghai, China.

Options	No. (%) of “Yes” in Total	No. (%) of “Yes” in Vaccinated Group	No. (%) of “Yes” in Unvaccinated Group	*p* Value for Chi-Squared Test
The influenza vaccination is the most effective measure to prevent influenza	2729 (66.7)	536 (76.0)	2193 (64.7)	<0.001
Aware of optimal time of vaccination	2104 (51.4)	412 (58.4)	1692 (49.9)	<0.001
People should be vaccinated every year	3025 (73.9)	592 (84.0)	2433 (71.8)	<0.001
Vaccinated healthcare workers (HCWs) can reduce the risk of infection in others	3622 (88.5)	645 (91.5)	2977 (87.8)	0.006
HCWs are the priority for vaccination	3050 (74.5)	568 (80.6)	2482 (73.2)	<0.001
Infants between 6 and 23 months of age are the priority for influenza vaccination	1777 (43.4)	324 (46.0)	1453 (42.9)	0.133
Family members and caregivers of infants under 6 months of age are the priority	1749 (42.7)	335 (47.5)	1414 (41.7)	0.005
Pregnant women are the priority for vaccination	1544 (37.7)	276 (39.2)	1268 (37.4)	0.387

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
