# Peer review of "An Exploratory Study of Influenza Vaccination Coverage in Healthcare Workers in a Western Chinese City, 2018–2019: Improving Target Population Coverage Based on Policy Interventions"

_vaccines, 2020, doi:10.3390/vaccines8010092_

Round 1

Reviewer 1 Report

These authors from the Institute for Infectious Disease Control and Prevention and Key Laboratory of Surveillance and Early-warning on Infectious Disease, Division of Infectious Disease, Chinese Center for Disease Control and Prevention, Beijing, present interesting findings her on Influenza vaccine coverage of health care workers in some of their large hospitals in Xining province.

Influenza vaccination is not mandatory among healthcare workers in Chinese healthcare settings and as a result, the coverage rate among HCW is very low (less than 15% since 2010). However, in 2017, a local, free influenza vaccine policy was implemented to boost coverage rate. The authors report that this policy has not been well implemented and the study data shows that many HCW still do  not know that they can obtain the vaccine for free at their local hospital site.

The current study presents data from 4 hospitals in the Qinghai province over a 1 year period from March 2018 to March 2019. Questionnaires were used to survey over 4000 HCW at these 4 sites, consisting mainly of younger workers (20-39 accounting for 83% of all workers surveyed). One third of those surveyed worked in pediatrics, ID, ED and respiratory departments.

Their data reveals that the coverage rates for the 2 hospitals without free vaccinations were 7.2% and 8.7% respectively, while that for the two hospitals that implemented this policy were 25.9% and 30.2% respectively. This is clearly a significant increase in coverage. It was interesting that of the respondents who did not get vaccinated, 25.6% did not know that the vaccine was free and 33.8% apparently did not realize that they could get it at their own facility! Also, the HCW who got vaccinated were significantly more likely to recommend the flu shot to their patients compared to the unvaccinated group (68.7% compared to 34.9%).

The manuscript presents interesting data and insights into the challenges of vaccination in the 21st century.

In general, the manuscript is well written with only minor sentence construction revisions needed.

Page 2 line 42 should read; “influenza whether or not they present with influenza-like symptoms, may…. The sentence that starts from line 41 to line 44 is too long and needs to be restructured into two separate sentences. Line 90 to 94: Again, this sentence is too long and should be two or three sentences in order to maintain the desired impact. Put a period after "season" (line 92) and start a new sentence there. Page 5, Figure 2B – the third label on the Y-axis should start with “Concerns about…” rather than “Concerned about”. On the same figure, the label for the orange bars should read “Numbers expressing willingness” instead of “Numbers of willingness” Page 6, Line 176, it is not clear what is meant by implementation of special funds here as an obstacle to vaccination among HCW. Please explain this to your readers. It appears that there is funding available to support this program but I cannot be sure of the nature of this. Page 6, Line 185; replace “vaccine uptake” with vaccination. Vaccine uptake might easily be read as dealing with cellular uptake of the vaccinate by antigen presenting cells. Line 192 should read; “…. participants was limited”.

Other minor changes

Page 1; Lines 23 and 24; You need a percentage here that represents the average coverage for hospitals that did not implement the free policy (2% and 8.7% respectively). Since this is the abstract, the data here should be as complete as possible for the reader to determine impact. Page 3, Line 91; you mention centralized or temporary vaccination sites here. In terms of proximity, where were these vaccinations being offered? Was this on the hospital grounds or did HCW have to travel to another facility to get vaccinated? A sentence or two about where in the hospital these vaccination sites are located would be very useful for readers.

General comment – Line 146 shows that 25.6% of HCW did not know that they could get vaccinated for free and the vast majority would have gotten the vaccine if they knew. How did this group not know that the vaccine was free? For the group that cited inconvenience as a reason for not getting the vaccine, were they just not aware that this service was available through a centralized or temporary facility?

Reviewer 2 Report

Thew maniscript lacks in novelty and in the methodology.

Concerning the last point, the authors present only descriptive analysis, bith in the abstract and in the Results section. However, no great inference can be derived from the paper due to the lack od a multivariate analysis.

I suggest to include in the revised version of the manuscript a more detailed analysis, using a multivariate analysis (logistic regression model).

Moreover, in the Discussion section I suggest to include some issue regarding the training for improving the vaccinazion rates. It is well known that the combination of an educational and a promotional element appared the most effective in augmenting the influenza vaccination coverage among health care workers. For details see

Schmidt S, Saulle R, Di Thiene D, Boccia A, La Torre G. Do the quality of the trials and the year of publication affect the efficacy of intervention to improve seasonal influenza vaccination among healthcare workers?: Results of a systematic review. Hum Vaccin Immunother. 2013 Feb;9(2):349-61

Round 2

Reviewer 2 Report

The authors made all the modifications required.

The manuscript is now suitabe for publication on the journal